# Forest Community Spatial Modeling Using Machine Learning and Remote Sensing Data

Artur Gafurov *[ID], Vadim Prokhorov, Maria Kozhevnikova and Bulat Usmanov[ID]

Institute of Environmental Sciences, Kazan Federal University, 5 Tovarisheskaya Str., 420097 Kazan, Russia; vadim.prokhorov@kpfu.ru (V.P.); mvkozhevnikova@kpfu.ru (M.K.); busmanof@kpfu.ru (B.U.)
* Correspondence: amgafurov@kpfu.ru

**Abstract:** This study examines the application of unsupervised classification techniques in the mapping of forest vegetation, aiming to align vegetation cover with the Braun-Blanquet classification system through remote sensing. By leveraging Landsat 8 and 9 satellite imagery and advanced clustering algorithms, specifically the Weka X-Means, this research addresses the challenge of minimizing researcher subjectivity in vegetation mapping. The methodology incorporates a two-step clustering approach to accurately classify forest communities, utilizing a comprehensive set of vegetation indices to distinguish between different types of forest ecosystems. The validation of the classification model relied on a detailed analysis of over 17,000 relevés from the "Flora" database, ensuring a high degree of accuracy in matching satellite-derived vegetation classes with field observations. The study's findings reveal the successful identification of 44 forest community types that was aggregated into seven classes of Braun-Blanquet classification system, demonstrating the efficacy of unsupervised classification in generating reliable vegetation maps. This work not only contributes to the advancement of remote sensing applications in ecological research, but also provides a valuable tool for natural resource management and conservation planning. The integration of unsupervised classification with the Braun-Blanquet system presents a novel approach to vegetation mapping, offering insights into ecological characteristics, and can be good starter point for sequestration potential of forest communities' assessment in the Republic of Tatarstan.

**Keywords:** unsupervised classification; vegetation mapping; forest communities; spatial modeling; remote sensing; multispectral imagery; Weka X-Means clustering algorithm; Braun-Blanquet classification system; Google Earth Engine; Landsat imagery

## 1. Introduction

Recently, renewed interest in vegetation classification has emerged worldwide, and efforts have been undertaken both nationally and internationally to develop new classification systems using standardized procedures [1,2]. Moreover, there is a growing interest in harmonizing approaches and standardizing the informational content of classifications [3]. This interest is driven by the practical necessity of using vegetation typologies and the increasing recognition of their scientific bases [4]. The classification problem is closely tied to assigning a particular community to a syntaxonomic unit of an existing vegetation classification. Expert systems have been used to solve this task [5]. The study from the *Journal of Plant Ecology* reviews the use of remote sensing imagery in vegetation mapping. It discusses the importance of selecting appropriate sensors based on spatial, temporal, spectral, and radiometric characteristics to map vegetation cover effectively. The study emphasizes the use of spectral radiances in the red and near-infrared regions, which can be incorporated into spectral vegetation indices (VIs) that are directly related to photosynthetic activity. The paper also mentions the use of airborne and space-borne sensors with spatial resolutions ranging from sub-meter to kilometers [6]. Another study, published in *Biodiversity and Conservation*, assesses vegetation mapping at different scales using two

common competing methods. The study focuses on the utility of these maps for reserve management. It mentions the use of high-resolution air photography and the importance of field reconnaissance. The study found that the relative accuracy of the Department of Environment and Conservation (DEC) and the CT method was 92% and 80%, respectively, when compared to local maps. The study also discusses the spatial accuracy of the Automated Photogrammetric Interpretation (API) reported as over 95% within 37.5 m for line work using higher-resolution (1:40,000) air photography [7]. The study from *Remote Sensing of Environment* proposes a method that links individual tree crowns (classified to species) using graph data structures and minimum spanning trees. The method involves clustering the delineated tree crowns, generating a graph data structure for each species, and then creating a minimum spanning tree using Kruskal's algorithm. The study implemented the method in C++ within the remote sensing and GIS software library using the Boost Graph Library [8]. The study in the *Forests* journal discusses forest cover mapping based on a combination of aerial images and satellite imagery. It highlights the data fusion of different spatial resolution remote sensing images applied to forest-type mapping. Although the specific clustering algorithm is not directly mentioned, the study likely involves advanced image processing and data fusion techniques to improve the accuracy of forest-type mapping [9].

Very recently, ideas have emerged regarding the amalgamation of vegetation classification procedures and the solutions provided by expert systems, employing the apparatus of fuzzy sets [10]. The proposed algorithms align with the notions of continuous vegetation cover and the individual behavior of plant species concerning environmental factors. Therefore, all these research directions aim to minimize the subjectivity of researchers at all stages of creating vegetation cover maps from the development of classification to assigning a specific geographical point to one type of ecosystem or another.

The analysis of terrestrial ecosystems, particularly the classification and mapping of vegetation, is a pivotal task for managing natural resources [11]. Flora forms the primary substrate upon which terrestrial life is established, exerting an essential influence across a multitude of ecological functions and environmental dynamics. In recent years, remote sensing technologies have substantially evolved, offering a robust tool for accurate and efficient vegetation mapping. As we delve into the realm of satellite imagery for the classification of forest communities, it becomes essential to clarify the scope and limitations of this approach. Satellite imagery, with its expansive coverage and high-frequency data acquisition, offers an unparalleled perspective for studying forest ecosystems. However, its efficacy in distinguishing between different tree species and, by extension, broader vegetation communities, is contingent upon the resolution and the type of data collected. Our study leverages this technology to focus primarily on tree species, which are identifiable with the current resolution of satellite data. The rationale behind this approach stems from the understanding that trees, as dominant components of forest ecosystems, serve as critical indicators of the overall health and composition of these ecosystems. A critical aspect of leveraging remote sensing data for vegetation mapping is the choice of classification approach, which largely dictates the accuracy and reliability of the resulting maps. Among the various classification approaches, unsupervised classification has emerged as a vital technique in minimizing researcher subjectivity at all stages of vegetation mapping, from developing a classification scheme to assigning a particular geographic point to a specific ecosystem type [12–16]. Unlike supervised classification, which relies on prior knowledge and extensive field samples, unsupervised classification operates without any a priori knowledge, solely depending on spectrally pixel-based statistics [13]. This inherent characteristic of unsupervised classification significantly reduces the potential bias that may arise from researchers' prior knowledge or assumptions, thereby enhancing the objectivity of the classification process. Unsupervised classification encompasses a variety of clustering algorithms, including rule-based, neural network, and SVM-based approaches, each with its unique advantages and applications in different scenarios. The utilization of unsupervised classification is not confined to preparing for supervised clas-

sifiers but extends to visualizing and monitoring similar areas in a scene, capturing a wide range of land use signatures over areas with extensive and unknown ranges of vegetation types. This broad capture capability underscores the versatility and efficacy of unsupervised classification in handling diverse ecosystems and varied vegetation types. Furthermore, the cost-effectiveness of unsupervised classification, in terms of both time and financial resources, is another significant advantage, especially when compared to supervised classification which necessitates a large number of field samples. The reduced need for extensive field sampling is particularly beneficial in scenarios such as agricultural production management, where timely access to spatial distribution information of crops is of great importance [6,17].

In light of the above, optimized unsupervised classification techniques have been proposed to further enhance the accuracy and efficiency of vegetation mapping. For instance, a two-step approach combining Jenks classification and Agglomerative Hierarchical Clustering (AHC) has been implemented to derive vegetation cover types from SAVI data [18]. Such advancements in unsupervised classification techniques are indicative of the ongoing efforts to minimize researcher subjectivity, thereby contributing to more accurate and reliable vegetation mapping.

The burgeoning interest and continuous advancements in unsupervised classification underscore its importance in the realm of vegetation mapping [6]. By minimizing researcher subjectivity, unsupervised classification paves the way for more objective, accurate, and reliable vegetation maps, which are indispensable for a myriad of ecological, environmental, and agricultural applications. The exploration and adoption of unsupervised classification techniques, coupled with the integration of other emerging technologies, hold promise for further advancing the field of vegetation mapping, fostering a deeper understanding of terrestrial ecosystems, and facilitating more informed decision-making in natural resource management [19].

The main goal of this study is to create a spatial model of forest vegetation in the classes of the Braun-Blanquet system. This classification approach is based on information about the composition and abundance of plant species in a community. Each species is characterized by its own functional characteristics, which ultimately can give a piece of knowledge about a particular community's productivity, which in turn can be used to judge the sequestration capacity of ecosystems.

When choosing a vegetation classification system, we opted for the Braun-Blanquet approach, as the entire community species composition is used to determine the vegetation class. This, along with the consideration of dominant species in the stand, reflects the ecological conditions of the habitat well [20,21].

Thus, ecological conditions on the one hand determine the vegetation classes in the Braun-Blanquet system, while on the other hand they can be identified by the remote sensing data. It is imperative to delineate here that our investigation does not delve directly into quantifying or analyzing carbon sequestration capabilities per se. Instead, it furnishes an essential precursor to such analyses. By advancing our understanding and methodological prowess in discerning and classifying forest vegetation with heightened accuracy, our research lays down a foundational bedrock. This groundwork is indispensable for subsequent, more focused inquiries into the carbon sequestration potential and dynamics inherent to various forest biomes.

This nuanced approach is not only reflective of our study's specific objectives but is also symptomatic of the contemporary ecological research ethos. In an era increasingly defined by climatic perturbations and the pressing need for environmental stewardship, our study contributes to the scaffolding required for constructing detailed, spatially informed ecological models. Such models are not just academic exercises but are pivotal tools in the arsenal against climate change, aiding in the formulation of responsive, informed, and sustainable environmental policies and practices.

## 2. Materials and Methods

A simplified representation of the work pipeline is shown in the flowchart in Figure 1.

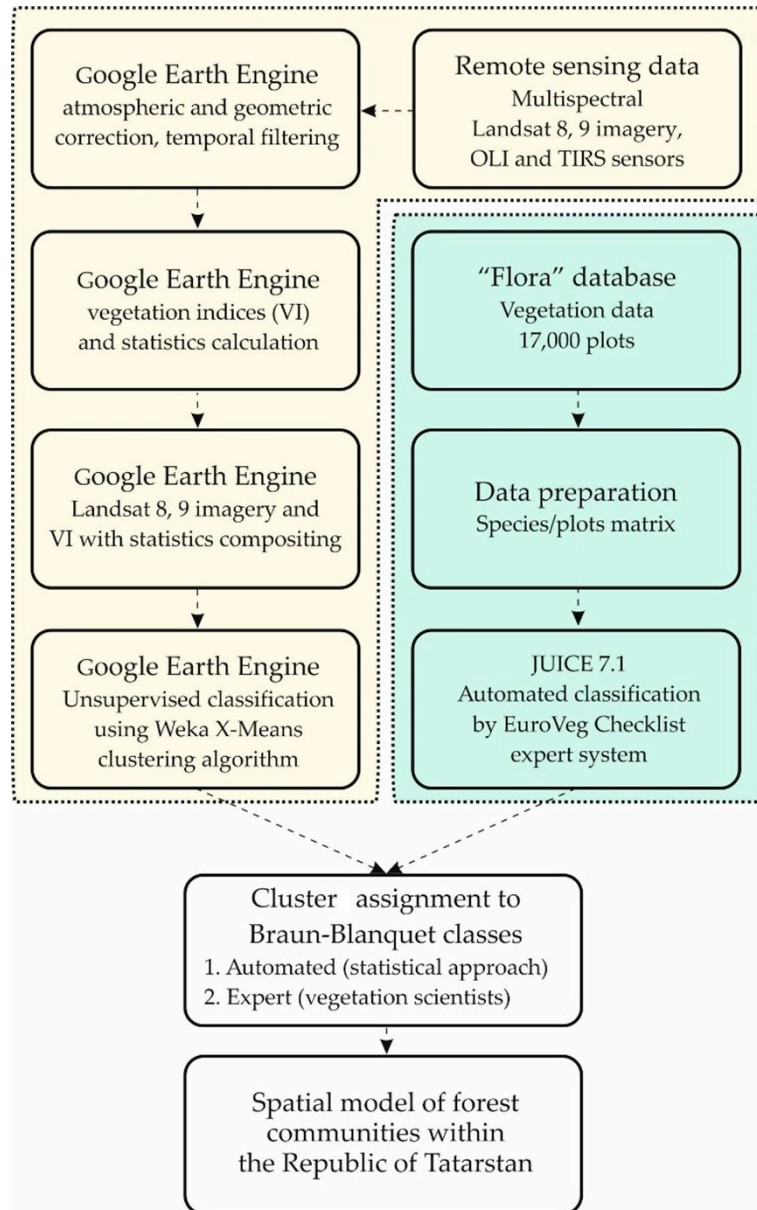

**Figure 1.** A simplified representation of the work pipeline.

### 2.1. Study Area

The study area is located within the Republic of Tatarstan (RT) (Russian Federation) situated in the eastern part of European Russia (Figure 2). The territory is divided by the river valleys of the Volga, Kama, and Vyatka Rivers into natural areas [22,23] including landscapes related to the Eastern European boreal subtaiga and Eastern semi-humid broadleaved forests; the boundary between them is determined by the isoline of hydrothermal coefficient of 1 [24]. In zonal terms, the northern part of the territory belongs to the ecoregion of Sarmatian mixed forests, and the southern part to the ecoregion of Eastern European forest-steppes [25].

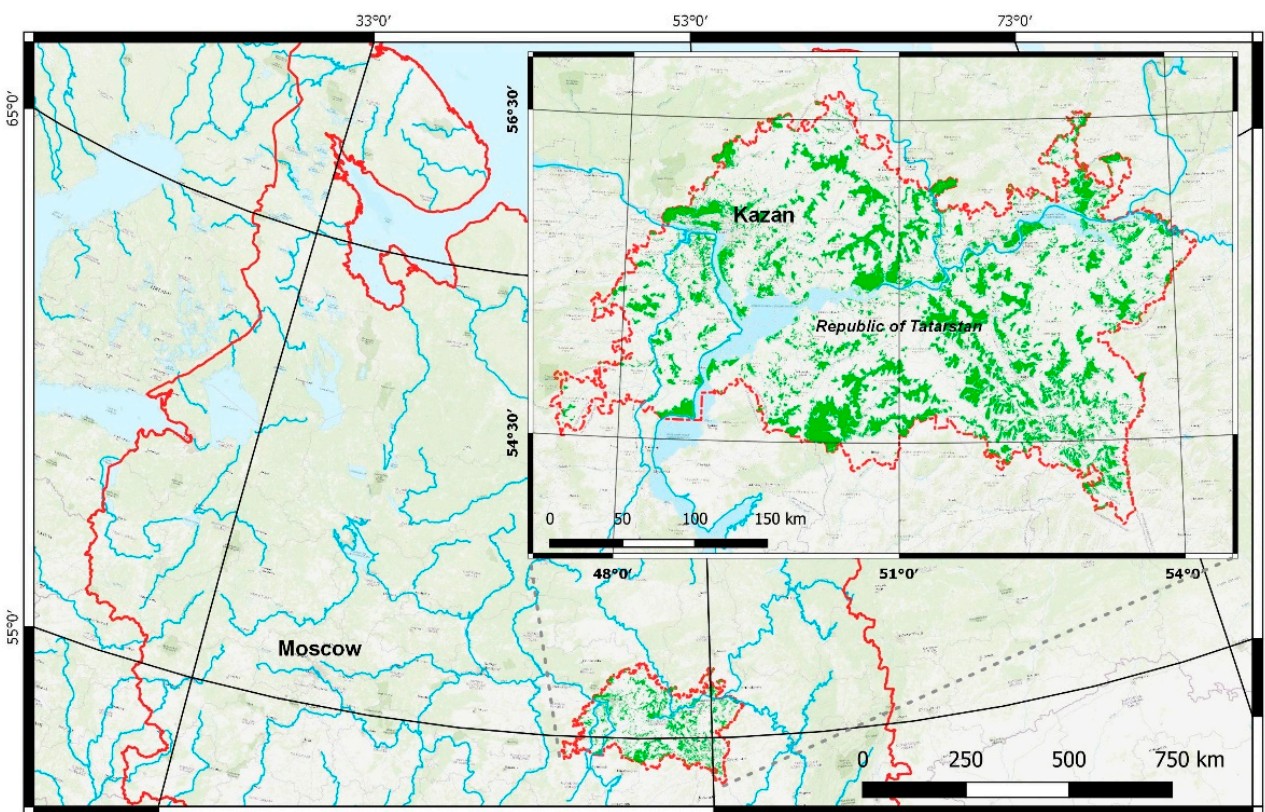

**Figure 2.** Study area. Forest distribution on the territory of the Republic of Tatarstan.

The area includes various landscape types and vegetation units ranging from the formation of the southern taiga to the fragments of typical steppes [26]. In the river valleys on sandy or sandy–clayey substrate, various types of pine forests are formed. Meadow formations have developed as a result of forest clearing and mowing. Steppes have remained mostly on steep rocky slopes that are not suitable for agricultural development. The predominant types of wetlands are fens.

The flora and vegetation of the Republic of Tatarstan have been studied mainly by researchers from Kazan University. This university has a long tradition of floristic and vegetation studies since the 19th century in connection with S. Korshinsky, A. Gordyagin, and M. Markov. Much phytosociological data have been collected for more than 120 years by various authors [27].

### 2.2. Field Data Collection and Processing

The primary source of data for this study's validation and verification was procured from the extensive "Flora" database [27], which houses a vast collection of over 17,000 relevés. Relevés, a phytosociological unit, provide a comprehensive description of plant communities. They present lists of species, detailing their abundance and stratification within the vegetation tier. The relevés in this database were conducted on plots of varying sizes, tailored to the type of vegetation community being surveyed. This adaptability in plot size is critical, as it ensures a representative sample of the vegetation is analyzed. For instance, herbaceous communities such as meadows and steppes were typically surveyed in smaller plots ranging from 1 square meter to 25 square meters, while larger plots up to 2500 square meters were used for forested landscapes to encompass their diverse tree species and structural complexity.

In this study, a selective extraction process was executed to acquire relevés documented post the year 2000, which also contained geographical references, ensuring relevance and contemporaneity of the data. For maintaining a standardized taxonomy across the dataset, the nomenclature of plant species was meticulously aligned with the globally recognized

"The Plant List" database [28,29]. This alignment is crucial to avoid discrepancies in species identification and ensure consistency in the dataset. A deliberate exclusion of bryophyte species from the selection was carried out to maintain a focused approach towards vascular plant communities. The classification of the relevés executed at the level of vegetation classes as delineated in the Braun-Blanquet system [30,31] provides a systematic method for vegetation analysis. This system, based on species presence and abundance, enables a detailed understanding of plant community structures and their ecological contexts.

Automation played a crucial role in the classification process. The expert systems EuroVeg Checklist [5] are employed within the JUICE 7.1 software package [32] to facilitate a precise and efficient automated classification. The integration of this advanced expert system streamlined the classification process, ensuring an objective and methodical analysis of the vegetation data. The use of such sophisticated tools underscores the importance of accuracy and efficiency in handling large datasets, characteristic of modern ecological studies.

### 2.3. Source Data Preparation

The source remote sensing data used were five years of Landsat 8 and 9 composite images for the period of April–November from 2018 to 2022. Adjusted images with processing level T1_L2—Surface Reflectance were used. All images for the selected period were processed for cloud and shadow removal. The resulting cleaned composites were used to calculate vegetation indices (Table 1) and their statistical metrics: median, mean, and maximum values, and standard deviation. To optimize data storage, each statistical metric was multiplied by 100 and converted to integer values.

The selected set of vegetation indices comprehensively describes the vegetation of the study area; however, in this study we focus on working only with forest species that have a large biomass. In order to analyze only forest species, a mask from the Global Land Analysis and Discovery (GLAD) laboratory at the University of Maryland [33] was used, effectively removing non-forested areas and thus honoring the focus on forested regions.

**Table 1.** Characteristics and applications of vegetation indices.

| Index Name | Description |
| --- | --- |
| Reflectance-based indices | |
| DVI (Difference Vegetation Index) [34] | Difference in reflectance between the near-infrared (NIR) and red bands. |
| DVIplus [35] | An extension of DVI, considering additional spectral bands or factors. |
| GNDVI (Green Normalized Difference Vegetation Index) [36] | Utilizes the green and NIR bands to emphasize chlorophyll content. |
| GRNDVI (Green-Red Normalized Difference Vegetation Index) [37] | Incorporates green and red bands for enhanced vegetation monitoring. |
| MNDVI (Modified NDVI) [38] | An adjusted form of NDVI to account for atmospheric and canopy background effects. |
| NDVI (Normalized Difference Vegetation Index) [39] | One of the most widely used indices, highlighting areas of active vegetation. |
| RDVI (Renormalized Difference Vegetation Index) [34] | Aims to minimize soil background influence while emphasizing vegetation. |
| TDVI (Transformed Difference Vegetation Index) [40] | A modified version of NDVI to enhance sensitivity to vegetation changes. |
| WDRVI (Wide Dynamic Range Vegetation Index) [41] | Offers a wider dynamic range than that of NDVI, improving sensitivity. |
| WDVI (Weighted Difference Vegetation Index) [42] | Focuses on minimizing soil background effects by applying a specific weight. |
| Water content indices | |
| DSWI1 (Drought Stress Water Index 1) [43] | Highlights areas undergoing drought stress, indicating lower water content. |
| GSAVI (Green Soil Adjusted Vegetation Index) [44] | A soil-adjusted index using the green band to minimize soil background influences. |
| NIRv (Near-Infrared Reflectance of Vegetation) [45] | Captures the NIR reflectance associated with vegetation's structural characteristics. |
| NIRvH2 [46] | A variation of NIRv, considering additional factors for enhanced accuracy. |
| NDII (Normalized Difference Infrared Index) [47] | Helps in assessing vegetation water content. |
| NDMI (Normalized Difference Moisture Index) [48] | Another index for evaluating water content in vegetation. |
| NMDI (Normalized Multi-band Drought Index) [49] | Focuses on monitoring drought conditions across various bands. |
| MSI (Moisture Stress Index) [50] | Highlights moisture stress in vegetation, crucial for drought monitoring. |

**Table 1.** *Cont.*

| Index Name | Description |
|---|---|
| | Soil-adjusted indices |
| GDVI (Green Difference Vegetation Index) [51] | Minimizes soil background influences using the green band. |
| GOSAVI (Green Optimized Soil Adjusted Vegetation Index) [44] | A soil-adjusted index designed for optimal vegetation monitoring. |
| GRVI (Green Red Vegetation Index) [44] | Utilizes green and red bands for vegetation analysis, adjusting for soil effects. |
| MSAVI (Modified Soil Adjusted Vegetation Index) [52] | A soil-adjusted index with modifications for better accuracy. |
| OSAVI (Optimized Soil Adjusted Vegetation Index) [53] | Optimizes soil adjustment to improve vegetation monitoring in low cover areas. |
| SARVI (Soil Adjusted and Atmospherically Resistant Vegetation Index) [54] | A soil-adjusted and atmospherically resistant index. |
| SAVI (Soil Adjusted Vegetation Index) [55] | Adjusts for the influence of soil bcenterness when vegetation cover is low. |
| TSAVI (Transformed Soil Adjusted Vegetation Index) [56] | A transformed index for better vegetation representation in diverse conditions. |
| | Chlorophyll indices |
| GARI (Green Atmospherically Resistant Vegetation Index) [36] | Designed to monitor chlorophyll content while minimizing atmospheric effects. |
| GCC (Green Chlorophyll Content) [57] | Directly related to chlorophyll content, crucial for assessing vegetation health. |
| MGRVI (Modified Green Red Vegetation Index) [58] | A modified index to enhance sensitivity to chlorophyll content. |
| MCARI2 (Modified Chlorophyll Absorption in Reflectance Index 2) [59] | Targets chlorophyll absorption features for accurate monitoring. |
| MSR (Modified Simple Ratio) [60] | A modified vegetation index to improve sensitivity to chlorophyll content. |
| MTVI2 (Modified Triangular Vegetation Index 2) [59] | Focuses on enhancing the representation of chlorophyll content. |
| OCVI (Optimized Chlorophyll Vegetation Index) [61] | Optimizes chlorophyll representation in vegetation monitoring. |
| VIG (Vegetation Index Green) [62] | Utilizes the green band for chlorophyll monitoring, essential for assessing plant health. |
| | Structural indices |
| CVI (Chlorophyll Vegetation Index) [63] | Represents vegetation structure and chlorophyll content. |
| DSI (Difference Structure Index) [64] | Highlights structural variations in vegetation. |
| SR (Simple Ratio) [65] | A basic ratio of NIR-to-red reflectance, indicating vegetation structure. |
| | Non-linear indices |
| EVI2 (Enhanced Vegetation Index 2) [66] | An improved version of NDVI, incorporating non-linear enhancements. |
| GBNDVI (Green Blue Normalized Difference Vegetation Index) [37] | A non-linear index utilizing green and blue bands. |
| GLI (Green Leaf Index) [67] | Represents vegetation greenness in a non-linear manner. |
| GEMI (Global Environment Monitoring Index) [68] | A global index for vegetation monitoring with non-linear properties. |
| RI (RapidEye Vegetation Index) [69] | A specific index for RapidEye satellite data. |
| SI (Shadow Index) [70] | Represents vegetation shape characteristics. |
| SEVI (Soil and Atmospherically Resistant Vegetation Index) [71] | Minimizes soil and atmospheric effects using a non-linear approach. |
| VARI (Visible Atmospherically Resistant Index) [62] | Focuses on the visible spectrum for vegetation monitoring, applying non-linear corrections. |
| | Other indices |
| BWDRVI (Broadband Width Difference Vegetation Index) [72] | A unique index capturing broadband width differences. |
| CIG (Canopy Index Green) [73] | Represents canopy structure using the green band. |
| FCVI (Floating Canopy Vegetation Index) [74] | A special index for floating canopy vegetation. |
| IAVI (Inverted Attributed Vegetation Index) [75] | An inverted index for enhanced vegetation attribute representation. |
| IKAW (Kawashima Vegetation Index) [76] | A specific vegetation index developed by Kawashima. |
| IPVI (Infrared Percentage Vegetation Index) [77] | Utilizes infrared reflectance for vegetation percentage estimation. |
| MRBVI (Modified Ratio Vegetation Index) [78] | A modified ratio index for improved vegetation monitoring. |
| MNLI (Modified Non-Linear Vegetation Index) [79] | Incorporates non-linear adjustments for enhanced vegetation representation. |
| NDDI (Normalized Difference Drought Index) [80] | Focuses on drought monitoring and water stress assessment. |
| NDGI (Normalized Difference Greenness Index) [35] | Highlights vegetation greenness. |

**Table 1.** *Cont.*

| Index Name | Description |
| --- | --- |
| NDPI (Normalized Difference Phenology Index) [81] | Utilized for monitoring vegetation phenology. |
| NDYI (Normalized Difference Yellowness Index) [82] | Highlights vegetation yellowness, crucial for certain phenological stages. |
| NGRDI (Normalized Green Red Difference Index) [83] | Utilizes green and red bands for vegetation monitoring. |
| NRFIg (Normalized Red/Far-Red Index Green) [84] | Focuses on the red-to-far-red ratio using the green band. |
| NRFIr (Normalized Red/Far-Red Index Red) [84] | Utilizes the red-to-far-red ratio in the red band. |
| NormG (Normalized Green) [44] | Represents normalized green reflectance. |
| NormNIR (Normalized NIR) [44] | Represents normalized near-infrared reflectance. |
| NormR (Normalized Red) [44] | Represents normalized red reflectance. |
| RCC (Red Chromatic Coordinate) [57] | Directly related to chlorophyll content using the red band. |
| RGBVI (Red Green Blue Vegetation Index) [58] | Utilizes the RGB bands for vegetation analysis. |
| RGRI (Red Green Ratio Index) [85] | A ratio index using red and green bands. |
| TGI (Triangular Greenness Index) [86] | Represents vegetation greenness using a triangular approach. |
| TriVI (Triangular Vegetation Index) [87] | A triangular index for enhanced vegetation representation. |

### 2.4. Remote-Sensing-Data-Based Clusterization

First, we generated a training dataset by randomly sampling 20,000 pixels from the prepared composite image within our masked region of interest (ROI). The training dataset was created with a scale of 30 m and consisted of 20,000 pixels to ensure adequate representation for each land cover class.

Next, we initiated the Weka X-Means clustering algorithm via the Google Earth Engine Clusterer object. By setting a minimum of 36 and a maximum of 50 clusters, we allowed the model to adaptively learn the optimal number of clusters based on the data. The trained clusterer was then applied to each sub-geometry in our region of interest.

To implement this, we wrote a user-defined function named "apply_clusterer", which took an input feature and retrieved the corresponding ROI. We clipped the prepared composite image based on the extracted ROI geometry and applied the trained clusterer model to generate clustered results. These results were tagged with their respective row_ids for further processing. For a fuller understanding and the reproducibility of the process, the source code is available on GitHub (https://github.com/noaakwey/geobotany/blob/main/Forest_communities_WekaXMean_Clustering.ipynb, accessed on 1 February 2024).

The clustered images were then compiled into an ImageCollection and transformed into a list of individual images using the toList() method. A mosaic image was generated from this list, allowing us to combine multiple clusters within our region of interest.

Lastly, we created a task configuration for exporting the resulting mosaic image as a GeoTIFF file to Google Drive.

The choice of setting the minimum cluster size at 36 and the maximum at 50 for our X-Means clustering algorithm is based on several factors. These settings are not arbitrary, but rather a balance between achieving accurate representation of various land cover classes and maintaining computational efficiency in large-scale land cover classification tasks.

Setting a minimum cluster size ensures that each identified land cover class has adequate representation within the dataset. A smaller minimum would risk underrepresentation, potentially leading to errors or misclassification. The specific value of 36 was chosen based on previous studies and research findings indicating that this number is sufficient for capturing the majority of distinct land cover classes in medium-sized regions.

On the other hand, setting a maximum cluster size limits the complexity of the clustering algorithm, which can help reduce computational costs and improve processing time. In our case, we set the maximum cluster size to 50 based on the assumption that the number of distinct land cover (forest) classes within our region would not exceed this limit. A larger "maximum" would increase the likelihood of encountering overlap-

ping or redundant clusters, which could lead to additional computational overhead and potential inaccuracies.

*2.5. Validation and Analysis*

The results of clustering are usually, in their raw form, a heavily interpreted dataset that must be reduced to the classes available in the study area. In our study, we developed a methodology to compare clustering results with the Braun-Blanquet classification of forest communities in the Republic of Tatarstan, supplemented by rigorous statistical analysis. The cluster model contained a mix of frequently and minimally occurring pixel values. To refine this model, a "tail cutting" approach was implemented, filtering out extreme percentile classes.

The next step involved correlating these raster model classes with the Braun-Blanquet classification. This process was executed using both automated and expert approaches. The automated method analyzed frequency tables of raster model values against vegetation plot coordinates. The expert method involved manual review of unallocated clusters by Braun-Blanquet classes based on available relevés. The aim was to identify classes with varying raster values, group these values based on Braun-Blanquet classifications, and then apply statistical analysis to establish significant associations.

This phase of aggregation led to the identification of principal community types, each corresponding to specific classes in the Braun-Blanquet classification. A Z-score assessment was integral to this stage, aiding in the normalization of data and establishing correlations between the raster model classes and the forest community types.

To validate our classification, the finalized raster model was integrated with an array of vegetation indices, followed by extensive statistical analysis. This involved categorizing the indices into relevant groups and applying various statistical tests, including the Kruskal–Wallis [88] and *t*-tests, supplemented by Cohen's d effect size [89] and the Dunn test [90]. Type I errors were controlled using the Bonferroni correction.

Significant disparities identified among the indices led to further analysis using the Mann–Whitney U-test [91] for pairwise comparisons between vegetation classes.

To evaluate the significance of each vegetation index and its corresponding statistical metric in the classification process, we employed a hybrid approach. The vegetation classes, already grouped according to the Braun-Blanquet classification system, served as reference values for training a supervised Random Forest model. It is important to note that this step was solely undertaken to quantify the contribution of each parameter to the final model and not for the purpose of classification itself.

The Random Forest algorithm, a powerful ensemble learning technique, was utilized for its ability to handle high-dimensional data and its robustness against overfitting. The model was trained using the vegetation indices and their statistical metrics (mean, median, maximum, and standard deviation) as predictor variables, while the target variable was represented by the vegetation classes derived from the Braun-Blanquet classification.

The training process involved the following steps:

1.  Data Preparation: The input data were partitioned into a training and testing set with a stratified sampling approach to ensure proportional representation of each vegetation class in both subsets.
2.  Model Training: The Random Forest algorithm was implemented using the scikit-learn library in Python. Hyperparameters such as the number of trees, maximum depth, and minimum samples per leaf were optimized through a grid search and cross-validation process.
3.  Feature Importance Evaluation: The trained Random Forest model provides a measure of feature importance, which quantifies the contribution of each predictor variable to the model's overall predictive performance. This information was used to assess the relative importance of the vegetation indices and their statistical metrics in distinguishing between the different vegetation classes.

In addition to the Random Forest analysis, the statistical significance of each parameter in the final classification was further evaluated using one-way analysis of variance (ANOVA). The ANOVA test was conducted to determine whether there were significant differences in the mean values of each vegetation index and its statistical metric across the different vegetation classes. This step provided complementary insights into the discriminatory power of the variables, supplementing the feature importance analysis from the Random Forest model. The results of the feature importance evaluation and ANOVA analysis were then used to identify the most influential vegetation indices and statistical metrics, enabling a better understanding of the key factors driving the classification of forest communities in the study area.

## 3. Results

As a result of the cluster analysis, 44 differing classes of forest communities were identified, which became the basis for the creation of a detailed raster model (Figure 3).

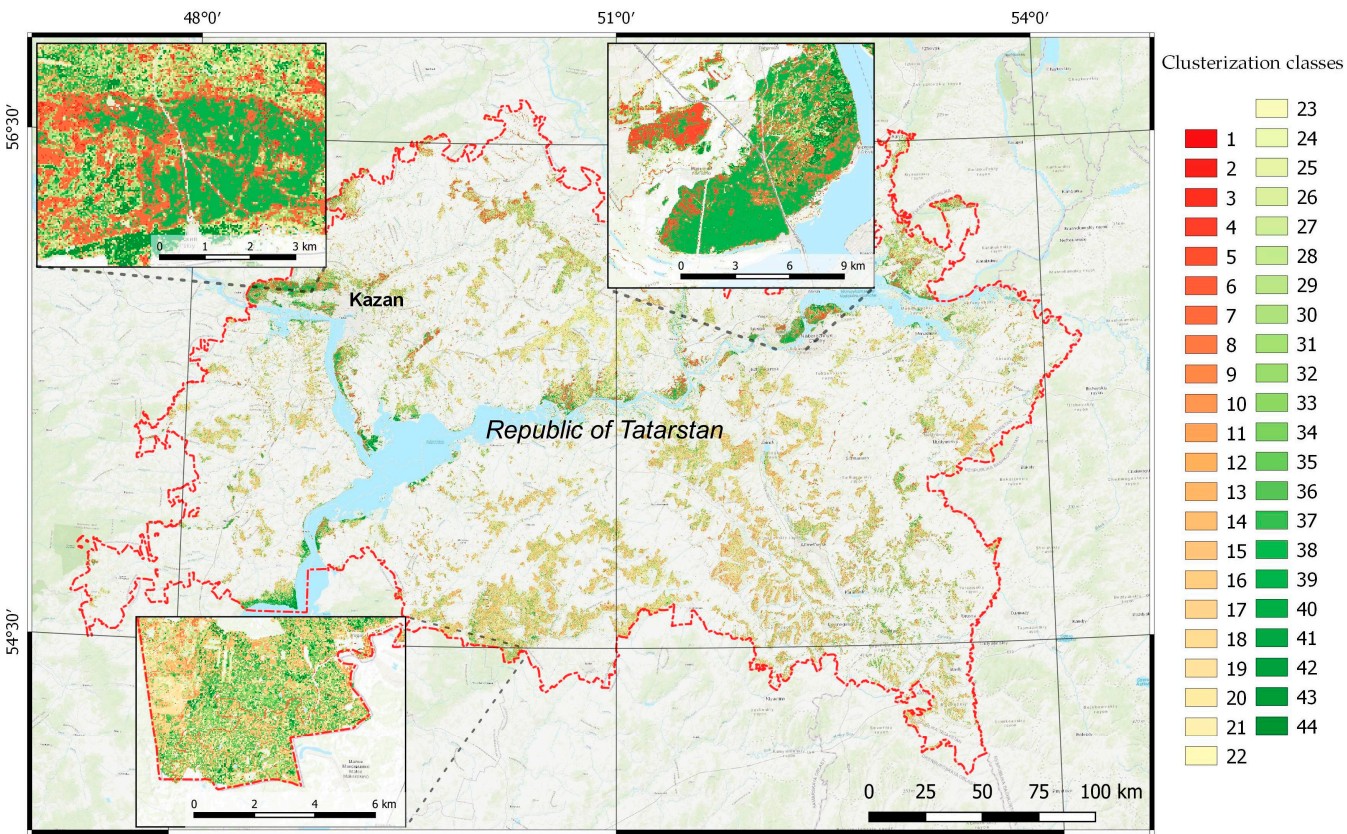

**Figure 3.** Spatial distribution of forest community model values.

The resulting model contains pixel values, some of which have minimal occurrence, providing a level of "noise" to the model, while others are frequently occurring values (Figure 4). Such "noisy" values need to be either excluded or associated with another class. To solve this problem, a "tail trimming" approach was used—at the 1st and 99th percentiles—to filter out all minimally occurring classes (values 3, 10, 34, and 40). Aggregation of the remaining image values of the clustering results (Figure 5) allowed us to assess consistency in the agreement between the predicted classes and the field data and allowed us to clearly identify the main community types, which are labeled as FAG (Carpino-Fagetea sylvaticae), BRA (Brachypodio pinnati-Betuletea pendulae), PIC (Vaccinio-Piceetea), ALN (Alnetea glutinosae), POP (Alno glutinosae-Populetea albae), and PUB (Quercetea pubescentis). The QUE (Quercetea robori-petraeae) class completely overlaps with the FAG group based on raster model values. After aggregation, a careful

Z-score was performed to measure the deviation of a particular observation from the mean value in the distribution. This measure is calculated as the ratio of the difference between the observation and the mean to the standard deviation. The Z-score is key to identifying deviant values, determining the percentiles of the data and the position of the observation within the confidence interval of the mean. It also facilitates normalization of the data, making it easier to analyze and compare the data. The remaining cluster model values show low statistical significance. These raster model values were not discarded but were allocated by expert judgment by visually comparing the constellations of unallocated pixel values and available relevés.

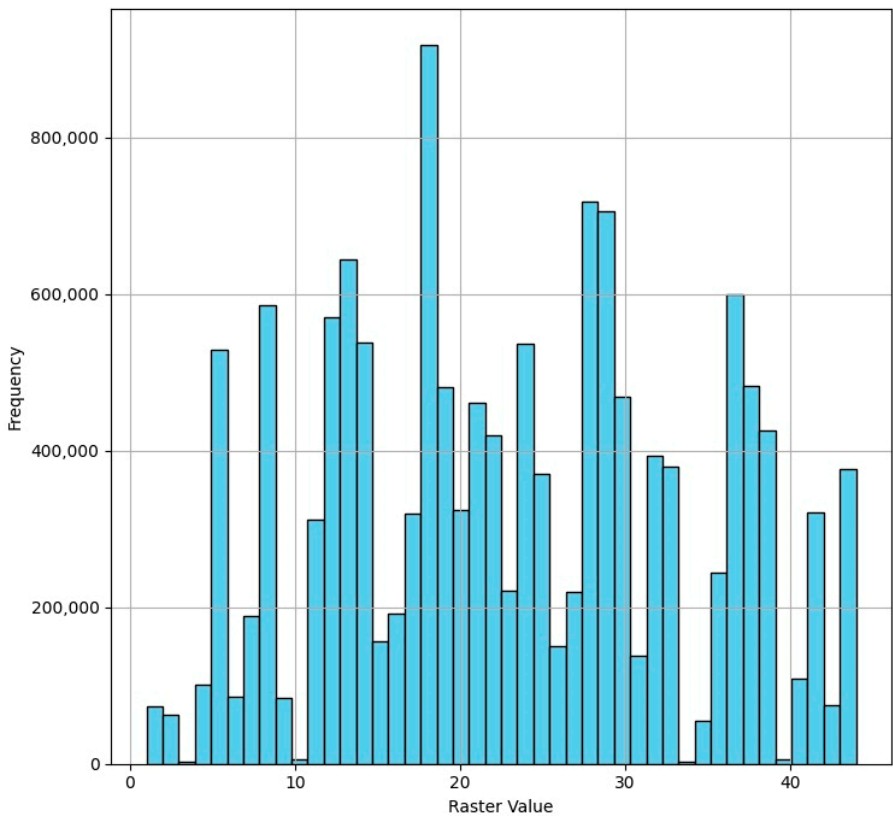

**Figure 4.** Histogram of raster values of the forest community model.

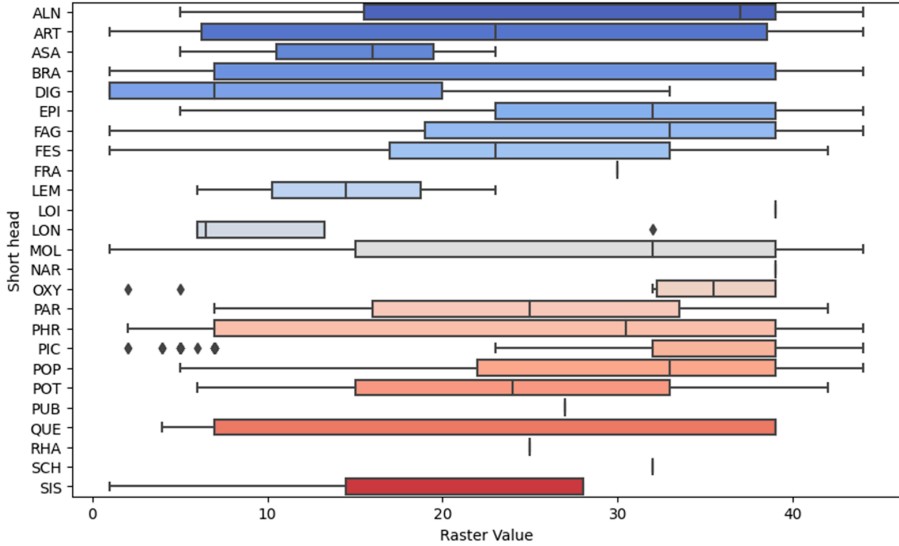

**Figure 5.** Aggregated raster values across field data classes (considering non-forest communities as well).

After all this manipulation, a reclassification table was created, based on which it became clear which clusters obtained from the results of the satellite images analysis correspond to Braun-Blanquet classes (Table 2).

**Table 2.** Braun-Blanquet classification and remote sensing classes comparison matrix.

| FC BB * | Raster Model Classes |
|---|---|
| ALN | 4 **, 5 **, 7 **, 17, 28, 42, 43 |
| BRA | 6 **, 36 ** |
| FAG | 11 **, 13, 14, 18, 19 **, 22 **, 23, 24, 29 **, 31, 34, 35, 37, 38, 44 ** |
| PIC | 2 **, 30 ** |
| POP | 1 **, 26, 33 **, 36 |
| PUB | 14 **, 25 **, 32 |
| OTHERS | *** |

* Forest communities' class by Braun-Blanquet. ** Raster model classes that could be clearly assigned to the corresponding FC BB class. *** All unallocated values were assigned to the OTHERS class.

It should be separately noted that not all values of the raster cluster model could be clearly assigned even with the expert approach in the analysis. Such groups of pixels either had an insufficient number of relevés or had a disputable character. In addition, as can be seen from Figure 5, some pixels of the cluster model fall in points with relevés of non-forest communities, which is due to errors in determining their coordinates reaching a value of 2 pixels. It was decided that all these values would be allocated to a separate OTHERS group and a separate field verification of these areas is planned for the future. In general, such areas correspond to no more than 13% of the entire forested territory of the Republic of Tatarstan or 1568 square kilometers (Table 3).

**Table 3.** Classified forest communities' raster model statistics.

| Class | Number of Pixels | Area (km$^2$) | % of Area of RT Forests |
|---|---|---|---|
| ALN | 2,082,534 | 1874 | 16 |
| BRA | 440,568 | 397 | 3 |
| FAG | 6,370,003 | 5733 | 48 |
| PIC | 564,690 | 508 | 4 |
| POP | 876,324 | 789 | 7 |
| PUB | 1,320,184 | 1188 | 10 |
| OTHERS | 1,742,692 | 1568 | 13 |

Thus, a modern map of the spatial distribution of forest communities in the Republic of Tatarstan according to the Braun-Blanquet classification was obtained (Figure 6).

To assess the involvement of selected predictor parameters in the final forest community model, a series of statistical tests were conducted. The basis was a hybrid approach based on training of the Random Forest supervised classification model, where all selected vegetation indices and their statistical metrics were used as predictors, and the variable was forest community classes according to the Braun-Blanquet classification (Tables S1–S7 in Supplementary Materials). This choice is due to the fact that Random Forest modeling allows the contribution of each factor in the final model to be explicitly estimated due to the Feature Importance property. This allowed us to calculate what percentage of the overall feature Importance is attributed to each of these parameters. This is helpful in understanding which features are the most significant to the model in predicting vegetation types.

The statistical description of the importance of the random forest model factors for classifying vegetation types provided several important insights. The range (1.31%) and standard deviation (0.17%) indicate significant variation in the importance of different factors. This shows that the contribution of different parameters to the model predictions varies considerably. The maximum importance value (1.32%) is significantly higher than

the mean (0.35%), indicating that there are several particularly important factors that have a significant impact on the model predictions. The median (0.30%) is lower than the mean (0.35%), which may indicate an asymmetric distribution of feature importance with a small number of very important features and a large number of less important features. Some factors are much more important for classification than others, which can be used to optimize and improve the model by focusing on the most important features. Overall, this suggests that there are both key factors and many less-influential factors in our model, which is common for multivariate datasets. Nevertheless, most of the factors have uniform non-zero importance, which indicates that the initial hypothesis in selecting a particular set of vegetation indices for training was correct.

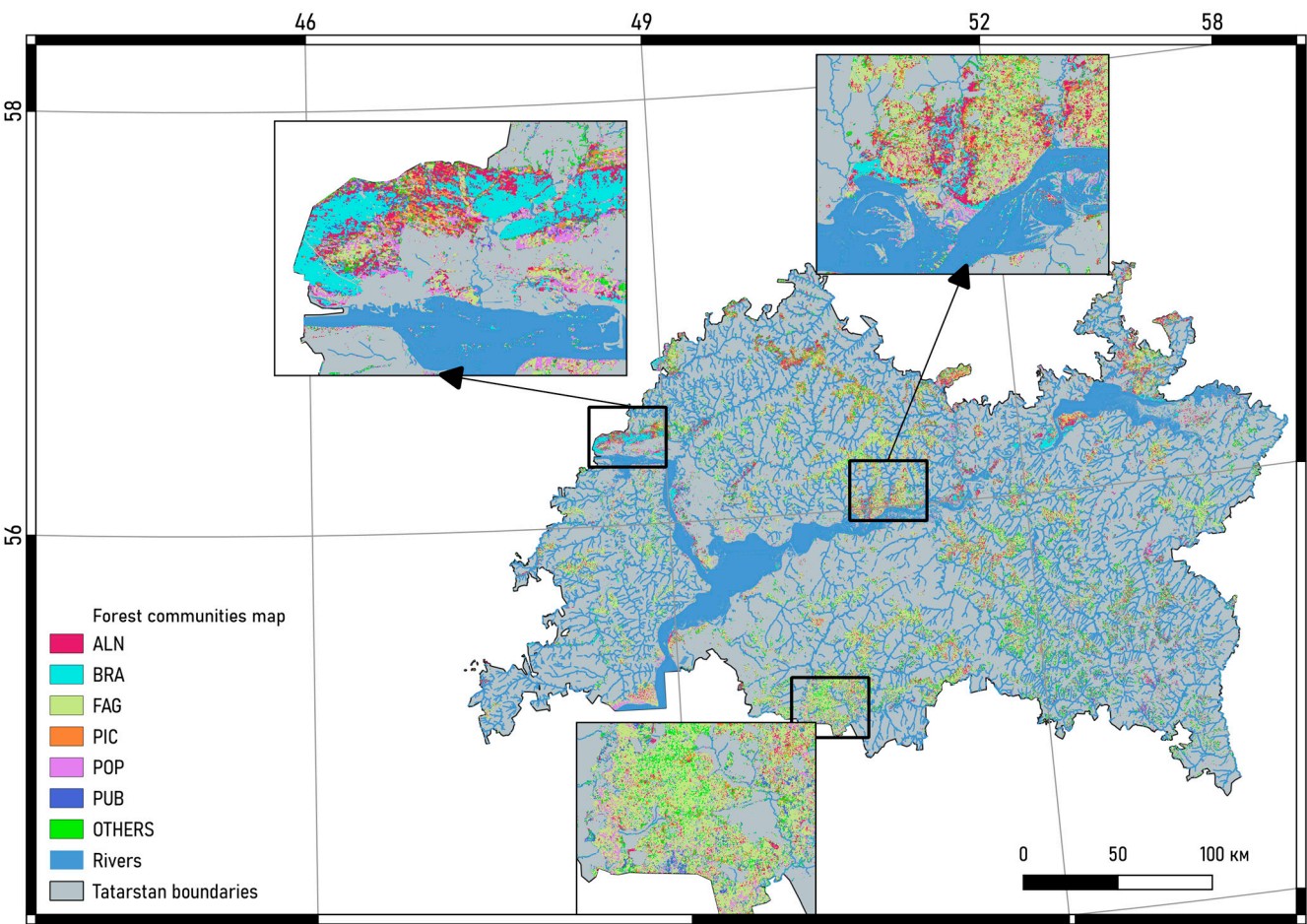

**Figure 6.** Result of spatial modeling of forest communities within the Republic of Tatarstan.

To find out which factors can be eliminated from the model without significant loss of accuracy, an experiment was conducted by removing the least important features step by step and evaluating the accuracy of the model after each removal. Training was stopped if the accuracy of the model dropped by more than 5% compared to the original model. The graph shows (Figure 7) that at a certain number of features, the accuracy of the model remains stable or even increases, after which it starts to decrease. The optimal number of attributes for the model, at which the maximum accuracy is achieved (about 62.5%), is 275; however, after iterative removal of the least significant attributes, the accuracy of the model begins to decrease by more than 5% of the maximum after removing nine attributes.

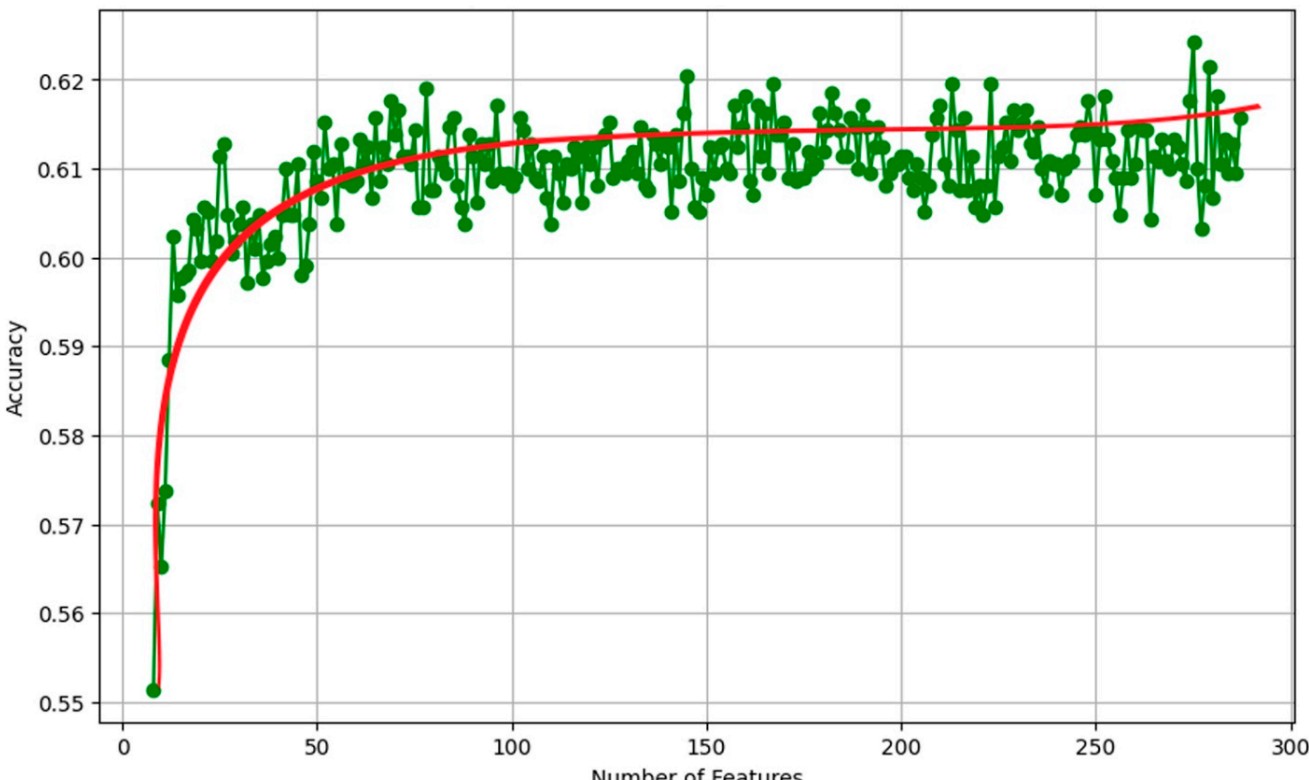

**Figure 7.** Relationship between the accuracy of the forest community classification model and the number of features (red line—is trend).

If we evaluate the obtained maximum accuracy, it is worth noting that the obtained accuracy is estimated as not very high; however, it should be noted here that this accuracy characterizes the model of aggregation up to seven classes of forest community types and not the initial results of unsupervised classification.

Additionally, the uniformity of feature involvement in the model was controlled using a cumulative curve. It shows how the importance of features accumulates as they are added. Each point on the graph corresponds to a feature sorted in descending order of importance (Figure 8).

ANOVA (Figure 9) and the non-parametric Kruskal–Wallis test were performed to evaluate the differences between groups (different vegetation types) on different indices. The results were aggregated into a single table sorted by importance and significance of the traits, which allowed ranking of all predictors (Table 4). These attributes are not only important for the model, but also have statistically significant differences between different vegetation types, making them particularly valuable for classification and analysis.

**Table 4.** Top 10 most important and representative features for classifying forest communities.

| Feature | Importance | F-Value | *p*-Value |
|---------|-----------|---------|-----------|
| TriVI_max | 1.319% | 1093.94 | <0.0000 |
| TriVI_stdDev | 1.175% | 1046.95 | <0.0000 |
| TGI_mean | 0.934% | 435.02 | <0.0000 |
| SEVI_median | 0.919% | 319.74 | <0.0000 |
| TGI_median | 0.871% | 541.51 | <0.0000 |
| FCVI_max | 0.870% | 1075.45 | <0.0000 |
| SR_median | 0.828% | 580.48 | <0.0000 |
| GRVI_median | 0.792% | 607.80 | <0.0000 |
| TriVI_mean | 0.748% | 910.28 | <0.0000 |
| NDYI_median | 0.732% | 202.37 | <0.0000 |

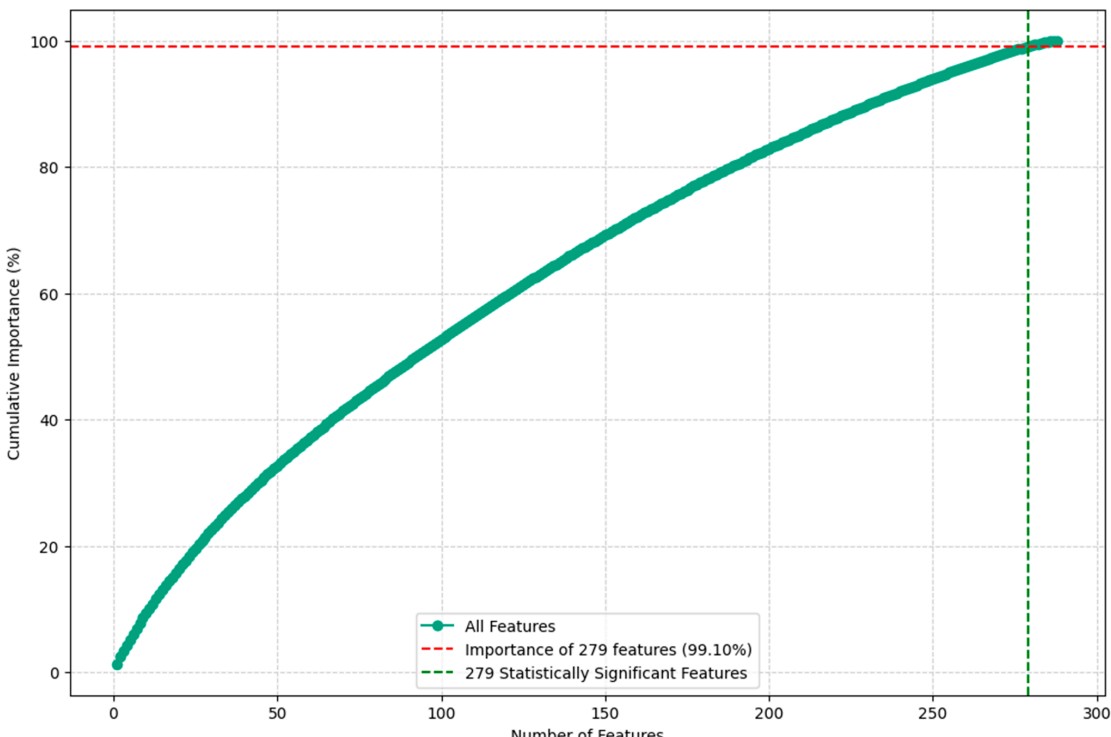

**Figure 8.** Cumulative importance curve of forest community model features.

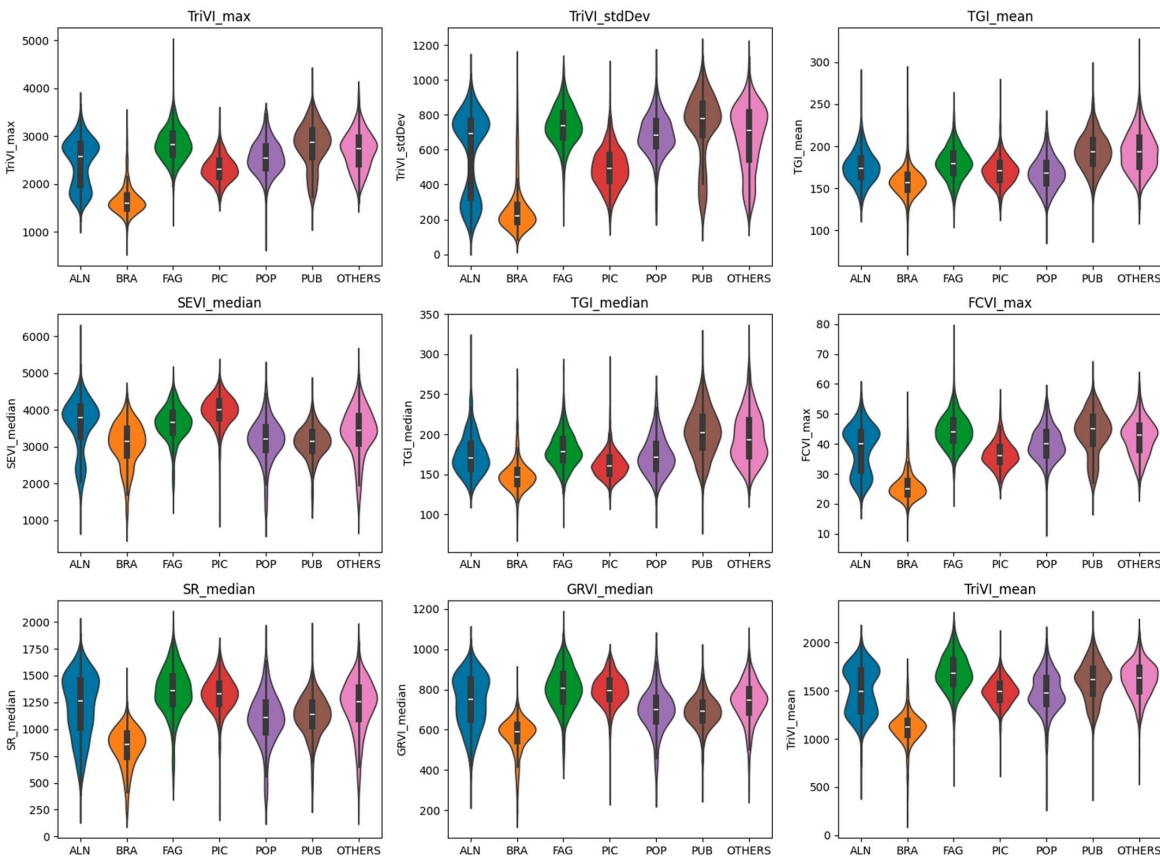

**Figure 9.** Violin charts of ANOVA test results. (Each "violin" reflects the density of the distribution of values of the trait, taking into account the type of vegetation. Wide parts of the "violin" indicate high density of values, and narrow parts indicate lower density).

## 4. Discussion

The analysis of average values of vegetation indices and statistical metrics aggregated across Braun-Blanquet classes allows for a comprehensive ecological description of each group.

In the rich mosaic of vegetation communities, those dominated by Alnus, especially prevalent in riparian zones, display a distinct ecological character molded by their hydrological surroundings, biological attributes, and notable vegetation indices. Alder trees, or Alnus, are commonly found thriving in wetland areas and along watercourses [92], well-suited to the moist conditions of these environments. The TriVI_max value of 24.49, while not exceptionally high, indicates a moderate to high canopy density that is characteristic of Alder communities. This canopy density is critical for various ecological functions, such as providing habitat and regulating the microclimate [93]. The standard deviation in TriVI, standing at 5.80, hints at some degree of variation within the canopy, possibly reflecting differences in the age and health of the trees or slight variances in species composition within these communities. A defining feature of Alders is their nitrogen-fixing capability, which significantly boosts soil fertility. This attribute is particularly valuable in often nutrient-scarce wetland environments and is essential for the sustenance of a wide array of flora and fauna that rely on these habitats. The DSI_max, at 1.18, suggests that these communities are generally resilient to severe drought stress, aligning with the Alders' preference for moisture-rich environments. The TGI_median and TGI_mean values, both at 1.75, serve as indicators of chlorophyll content, pointing to a robust photosynthetic activity crucial for the growth and health of these communities. This robust photosynthesis is further evidenced by an EVI2_max value of 0.66, signaling healthy vegetation. The MSR_mean, at 2.67, and MSI_max, at 1.18, indicate a well-balanced moisture regime within the foliage, underscoring the Alders' adaptation to wet conditions. The NDYI_median of 0.36 and FCVI_max of 0.38, associated with foliage conditions, suggest active photosynthetic tissue, essential for maintaining the productivity and ecological equilibrium of riparian zones [94]. The GEMI_max of 0.92, indicative of photosynthetic efficiency, and the GRVI_median of 7.43, related to vegetation greenness, further reinforce the vitality and health of these communities. Hydrologically, Alders significantly impact riparian and wetland ecosystems. Their presence typically indicates high soil moisture levels, a crucial element in the hydrology of these areas. This moisture not only benefits the Alders but also fosters an environment conducive to a variety of other plant and animal species, thereby enhancing overall biodiversity. The dense canopies of Alder communities provide vital habitats and nesting sites for numerous bird species, while their root systems play a supportive role in sustaining aquatic life in nearby water bodies [95].

The Brachypodio pinnati-Betuletea pendulae communities, marked by a unique blend of Brachypodium pinnatum grasses and Betula pendula birch trees, exhibit a distinctive ecological and botanical landscape. These communities often emerge in transitional ecological zones, forming a rich tapestry of interactions between various plant species and their environment [96]. Characterized by moderate canopy density as indicated by a TriVI_max of 16.65, these ecosystems present a varied habitat where open grasslands merge with denser birch canopies. This heterogeneity is essential for the coexistence of diverse species, supporting a range of ecological functions from habitat provision to nutrient cycling [97]. The TriVI_stdDev of 2.68 underscores the variability within these communities, likely due to the intermingling of different vegetation types. Ecologically, these communities are resilient to drought, as suggested by a DSI_max of 1.21. This resilience is attributable to the adaptive strategies of both the grasses and birch trees, which are accustomed to varying moisture conditions. Efficient photosynthesis is evident from the TGI_median and TGI_mean values of 1.50 and 1.58, respectively, highlighting the robust chlorophyll content necessary for the growth and health of these plants. The moisture regime within the foliage, represented by an MSR_mean of 2.21 and MSI_max of 1.21, points to a balanced moisture content, reflective of the diverse moisture needs within these mixed communities. The GEMI_max of 1.10 and NDYI_median of 0.34 further affirm the healthy state of photosynthetic tissues

among the varied plant species [97]. Topographically, these communities often signify early successional stages or areas recovering from disturbance, marking a landscape in transition. This transitional nature is pivotal in maintaining biodiversity and ecosystem dynamics. Indicators of healthy vegetation, such as a moderate EVI2_max of 0.49 and MCARI2_max of 0.49, are critical for soil stabilization and maintaining the integrity of these evolving landscapes. Hydrologically, the MSR_median of 2.37 and DSI_stdDev of 0.17 imply a stable water regime with minimal fluctuation in drought stress, underscoring the importance of water management in these areas. The SR_median of 8.32 and GRVI_median of 5.72, linked to the canopy's reflectance properties, influence the hydrological dynamics, contributing to leaf area and overall vegetation health [98].

Fagus-dominated communities, primarily comprising beech forests, stand out with their distinct ecological and botanical characteristics, deeply ingrained in their environmental settings and species-specific traits [99]. These communities, emblematic of climax ecosystems in temperate zones, are defined by dense, shade-tolerant canopies that foster a unique microclimate beneath them. The vegetation indices, including TriVI_max (28.31), DSI_max (1.43), and SEVI_median (36.24), provide detailed insights into these characteristics, underpinning the complex ecological dynamics of these forests. The TriVI_max of 28.31, while not exceptionally high, indicates substantial canopy density typical of mature beech forests. This density is critical in stabilizing the ecosystem, fostering a stable and resilient environment conducive to a variety of understory species. The TriVI_stdDev of 7.38 points to canopy variability, potentially reflective of different forest development stages or microclimatic variations within the forest [100]. This variability is essential in maintaining the forest's ecological diversity and adaptability. The moderate DSI_max of 1.43 aligns with the beech trees' ability to adapt to varying moisture regimes, highlighting their ecological resilience. The TGI_median (2.03) and TGI_mean (1.94) suggests a healthy vegetation state, indicative of robust photosynthetic activity crucial for the shade-tolerant canopies. The MSR_mean of 2.49 and MSI_max of 1.43 indicate a balanced moisture content in the foliage, vital for the health and growth of beech trees [101]. These indices reflect the intricate balance between moisture availability and the physiological needs of the beech forests. Beech forests, as climax communities, signify ecosystem stability and maturity. They play a pivotal role in shaping the understory's microclimate and supporting a wide range of flora and fauna. The dense canopies provide critical ecological services like carbon sequestration and habitat provision. The NDYI_median of 0.36 and FCVI_max of 0.44, associated with active photosynthetic tissue, further support the forests' significant ecological roles. Topographically, beech forests commonly inhabit hilly or mountainous terrains [102]. The EVI2_max (0.71) and MCARI2_max (0.76) indicates healthy vegetation, essential in maintaining the topographical integrity of these regions. The GEMI_max of 1.11 and GRVI_median of 7.38 reinforce the health and vitality of these communities, underscoring their importance in the landscape. Hydrologically, these forests significantly influence the local water cycle. The MSR_median of 3.06 illustrates their contribution to the area's moisture regime, a critical aspect in temperate forest ecosystems. The low DSI_stdDev of 0.28 indicates stable drought stress conditions, ensuring consistent hydrological functioning within the forests.

Picea, or spruce-dominated communities, represent a unique ecological composition characterized by a rich interplay of botanical characteristics, topographical features, and hydrological aspects. These communities, vital to both boreal and montane ecosystems, are defined by dense, coniferous canopies typical of spruce forests, which play a significant role in the ecosystem's structural and functional dynamics [103]. The vegetation indices, including TriVI_max (23.24), DSI_max (0.98), and SEVI_median (31.70), offer deep insights into these dynamics, reflecting the health and robust state of these coniferous vegetations. The TriVI_max of 23.24, indicative of a moderately high canopy density, highlights the dense stands characteristic of spruce forests. This canopy density is crucial for ecological functions like habitat provision and microclimate regulation. The TriVI_stdDev of 4.97 suggests variability within the canopy, likely due to differences in tree age, health, or

species composition, which is crucial for the ecological diversity and adaptability of these forests. Spruce trees exhibit significant ecological resilience, demonstrated by a DSI_max of 0.98, pointing to their lower susceptibility to drought and adaptability to various moisture regimes. Spruce forests thrive in cooler climates and a range of soil conditions, from dry, sandy soils to moist environments. In drier climates, their ability to grow in less moist conditions underscores their ecological versatility. These forests contribute significantly to soil stabilization, preventing erosion, and influencing soil properties such as moisture retention, soil pH, and organic matter content [104]. Their needle litter plays a key role in shaping the soil composition, impacting the understory vegetation and soil microbial communities. The TGI_median of 1.62 and TGI_mean of 1.71, indicators of chlorophyll content, further affirm the efficient photosynthetic activity within these forests. The MSR_mean of 2.82 and MSI_max of 0.98 indicate a balanced moisture content in the foliage, essential for the health and growth of spruce trees. The GEMI_max of 0.95 and NDYI_median of 0.31 reinforce the healthy state of photosynthetic tissues. Topographically, spruce forests often occupy hilly or mountainous regions, contributing to the ecological integrity of these landscapes [105]. The moderate values of EVI2_max (0.64) and MCARI2_max (0.72) indicate healthy vegetation, vital for maintaining the landscape's topographical integrity. The FCVI_max of 0.36 and GRVI_median of 6.90 underscore the foliage conditions and photosynthetic capacity of spruce trees, influenced by their topographical positioning [106]. Hydrologically, spruce forests play a significant role in their ecosystems, contributing to the moisture regime of the area, essential in both boreal and montane environments. The MSR_median of 2.92 indicates their contribution to the area's moisture regime, and the low DSI_stdDev of 0.18 suggests stable water stress conditions, ensuring consistent hydrological functioning. The SR_median of 11.27 and NIRvH2_max of 0.37 provide insights into the canopy's reflectance properties, linked to leaf area and overall vegetation health, influencing the hydrological dynamics of the ecosystem [107].

Populus, or poplar-dominated communities, exhibit a unique ecological identity deeply rooted in their botanical characteristics, topographical diversity, and hydrological aspects. These fast-growing trees, predominantly found in riparian zones, play critical roles in ecosystem dynamics, with their dense and rapid growth reflected in various vegetation indices like TriVI_max (25.64), DSI_max (1.45), and SEVI_median (31.70). The TriVI_max of 25.64 indicates a relatively high canopy density characteristic of poplar trees, integral to the ecological functioning of riparian zones [108]. They provide vital habitat and significantly influence the microclimate. The TriVI_stdDev of 6.85 points to variability within these communities, likely due to differences in tree age or environmental conditions [109]. Poplars are known for their remarkable adaptability, particularly in riparian environments where they quickly colonize and stabilize the habitat. This adaptability is evident in their moderate level of drought tolerance, as suggested by a DSI_max of 1.45, suitable for their fluctuating moisture conditions. The TGI_median of 1.74 and TGI_mean of 1.68, indicators of chlorophyll content, suggest efficient photosynthesis, critical for the growth and health of these communities [110]. The MSR_mean of 2.45 and MSI_max of 1.45 denote a balanced moisture regime within the foliage, reflecting the diverse water requirements of poplar trees. The GEMI_max of 0.99 and NDYI_median of 0.31 further reinforce the healthy state of photosynthetic tissues in these trees. Topographically, poplar communities are crucial in stabilizing riverbanks and streams, with healthy vegetation indicated by moderate values of EVI2_max (0.68) and MCARI2_max (0.72). The FCVI_max of 0.40 and GRVI_median of 6.95 underscore the foliage conditions and photosynthetic capacity of the poplars, influenced by their topographical positioning. Hydrologically, poplar forests significantly impact the water dynamics of riparian zones [111]. The MSR_median of 2.85 suggests their role in the area's moisture regime, essential in floodplain ecosystems. The low DSI_stdDev of 0.30 indicates stable water stress conditions, ensuring consistent hydrological functioning. The SR_median of 10.94 and NIRvH2_max of 0.41 provide insights into the canopy's reflectance properties, linked to leaf area and overall vegetation health, influencing the hydrological dynamics of the ecosystem.

*Quercus pubescens* (or pubescent oak)-dominated communities, present a unique ecological, botanical, topographical, and hydrological profile, distinguishing them significantly within various ecosystems. These communities, typically found in well-drained, often calcareous soils in warm, dry climates, showcase a distinct interplay of environmental factors that shape a unique ecological niche [112]. The vegetation indices for these communities, such as TriVI_max (28.01) and EVI2_max (0.72), reflect dense and healthy canopies that are characteristic of oak woodlands. The TriVI_max of 28.01 indicates a relatively high canopy density, crucial for the ecological functioning of pubescent oak communities, providing habitat and regulating the microclimate. The TriVI_stdDev of 7.36 highlights the variability within the canopy, possibly due to differences in tree age, health, or environmental conditions. The DSI_max of 1.40 suggests moderate drought tolerance, emblematic of the adaptability of pubescent oaks to typically dry environments. Pubescent oaks are known for their adaptability to various soil types, including less fertile soils [113]. This adaptability is echoed in their diverse range of vegetation indices. The TGI_median of 2.03 and TGI_mean of 1.94, indicating chlorophyll content, suggest efficient photosynthetic activity, essential for maintaining the health and productivity of these trees. The MSR_mean of 2.49 and MSI_max of 1.40 indicate a balanced moisture regime in the foliage, crucial for the growth of these oaks in drier climates. Topographically, these communities are often found in hilly or mountainous regions. The moderate values of EVI2_max and MCARI2_max (0.76) suggest healthy vegetation, vital for maintaining the topographical integrity of these areas. The NDYI_median of 0.38 and FCVI_max of 0.44 highlight the foliage conditions and photosynthetic capacity of the oaks, influenced by the topographical features of their habitat [114]. Hydrologically, pubescent oak communities play a significant role in their ecosystems. Their root systems are adept at tapping into deeper water sources, essential in drier environments. This ability not only supports the trees themselves but also affects the surrounding vegetation and overall ecosystem water balance [115]. The MSR_median of 2.92 and the low DSI_stdDev of 0.30 suggest stable water stress conditions, supporting consistent hydrological functioning. The leaf litter and soil organic matter under these oaks contribute to water retention and soil health, enhancing the overall hydrological functioning of the area [116].

The category "OTHERS" in vegetation community classification includes a diverse range of ecosystems that defy conventional categorization into specific vegetation types. This diversity, which encompasses a variety of species and ecological interactions, is evident in the vegetation indices such as TriVI_max (27.00), DSI_max (1.28), and SEVI_median (33.98). These indices offer a glimpse into the intricate ecological fabric of these mixed communities. The TriVI_max of 27.00 indicates a high canopy density, suggesting the presence of densely vegetated areas within these mixed communities. This density likely results from the amalgamation of various vegetation types, each contributing uniquely to the ecosystem's overall structure and function. The TriVI_stdDev of 6.71 highlights the variability in canopy density, characteristic of mixed communities where diverse species coexist, each with its unique growth patterns and structural traits. Ecologically, the "OTHERS" category likely comprises a mix of vegetation types, possibly representing transitional zones or ecotones, areas where different ecological communities converge. This diversity leads to high biodiversity and ecological productivity, making these areas crucial for regional biodiversity maintenance. The moderate DSI_max of 1.28 suggests a reasonable degree of drought resilience, likely owing to the varied adaptive strategies of the constituent species. The SEVI_median of 33.98, reflecting soil influence on vegetation signals, underscores the complex interactions between different plant types and the soil environment. Botanically, this category includes a diverse range of plant species, each with its adaptations to the local environment. This could range from drought-resistant species in dryer areas to water-loving plants in wetter parts of the community. The MSR_mean of 2.64 and MSI_max of 1.28 indicate a balanced moisture regime, essential for the diverse growth requirements within these communities. The TGI_median of 1.96 and TGI_mean of 1.94, indicative of chlorophyll content, suggest efficient photosynthetic activity across

the spectrum of species, critical for the overall productivity and health of the ecosystem. Topographically, these mixed communities can be found across varied landscapes. The EVI2_max of 0.71 and MCARI2_max of 0.75, signaling healthy vegetation, are vital for maintaining the integrity of these diverse landscapes. The NDYI_median of 0.38 and FCVI_max of 0.42 further underscore the foliage conditions and photosynthetic capacity influenced by the topographical features of their habitats. Hydrologically, the varied vegetation types within the "OTHERS" category have diverse impacts on local water cycles. The MSR_median of 3.06 suggests a significant role in the moisture regime, essential for maintaining the hydrological balance in these ecosystems. The low DSI_stdDev of 0.25 indicates stable water stress conditions across the communities. The SR_median of 12.17 and NIRvH2_max of 0.43 provide insights into the canopy's reflectance properties, linked to leaf area and overall vegetation health, influencing the hydrological dynamics of the ecosystem.

## 5. Conclusions

The results of the study showed that through unsupervised classification of multispectral satellite images, areas of vegetation cover that correspond with high confidence to vegetation classes in the Braun-Blanquet system are identified. Based on the obtained model and data on the sequestration potential of various types of forest communities, future studies are planned to determine the carbon stock of ecosystems of the Republic of Tatarstan. The vegetation changes over time and space as a result of ecological processes acting on plant populations and communities at different temporal and spatial scales, making it one of the integral indicators of the ecosystem as a whole [117]. This is because the vegetation component of ecosystems is formed depending on both abiotic factors (moisture, light, and soil fertility), landscape characteristics (e.g., topography and drainage) [118,119], and anthropogenic influences. In addition, the amount of carbon stored in soil and wood depends on the successional status of the community (tree age and crown closure) [120–122]. We hypothesize that the spectral characteristics of satellite imagery using different indices reflect the above factors [20]. Hence, we propose utilizing the spatial model of vegetation cover as a basis for evaluating the sequestration potential of ecosystems.

Our investigation, while primarily pivoting around the utilization of remote sensing technologies for the robust classification of these forest communities, acknowledges its tangential yet significant contribution to the broader landscape of carbon balance studies.

These data can also serve as a cartographic base for planning and implementing climate projects.

**Supplementary Materials:** The following supporting information can be downloaded at: www.mdpi.com/xxx/s1, Tables S1–S7: Resulted vegetation metrics on 7 forest community classes.

**Author Contributions:** Conceptualization, V.P. and M.K.; methodology, A.G.; software, A.G.; validation, A.G. and V.P.; formal analysis, A.G.; investigation, V.P. and M.K.; writing—original draft preparation, A.G.; writing—review and editing, B.U.; visualization, A.G. and B.U.; supervision, V.P.; project administration, M.K.; funding acquisition, B.U. All authors have read and agreed to the published version of the manuscript.

**Funding:** This work was funded by the subsidy allocated to Kazan Federal University for the state assignment in the sphere of scientific activities, project No. FZSM-2024-0004.

**Data Availability Statement:** The original contributions presented in the study are included in the article and Supplementary Materials, further inquiries can be directed to the corresponding author.

**Conflicts of Interest:** The authors declare no conflicts of interest.

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
