# Peer review of "Forest Community Spatial Modeling Using Machine Learning and Remote Sensing Data"

_remotesensing, doi:10.3390/rs16081371_

Round 1

Reviewer 1 Report

Comments and Suggestions for Authors

The submitted manuscript delves into a methodology for mapping forest communities in the Republic of Tatarstan using unsupervised classification of Landsat images alongside a field species database. The WekaXMeans classifier was applied to an asset comprising Landsat images and spectral indices, resulting in the identification of 44 clusters that were then linked to dominant forest communities, culminating in the delineation of seven distinct classes.

The paper boasts a well-structured and scientifically rigorous framework, employing statistical techniques for validation and accuracy assessment. It also effectively engages with previous literature, particularly in the discussion and interpretation of results. However, despite its comprehensive nature, there are several areas that require clarification. For instance, Figure 5 displays approximately 25 field data classes, yet post-aggregation, only seven classes (referred to as forest communities) remain, without a clear explanation of the criteria used to define them as primary categories. Additionally, the utilization of Random Forest in the validation process warrants a more detailed description within the methodology section.

While the results presented demonstrate consistency and robustness, the necessity for manual adjustments hinders the widespread applicability of the method. Efforts to address these gaps and provide clearer explanations would significantly enhance the overall quality and comprehensibility of the manuscript.

Author Response

Dear Reviewer,

Thank you for the insightful comments and constructive feedback on our manuscript. We greatly appreciate the time and effort invested in providing a thorough review, as it helps us enhance the quality and clarity of our work. Please allow us to address some of the points raised:

Regarding the aggregation of the clusters into seven major forest communities: the primary criterion for combining individual clusters into larger groups was their correspondence to field data from the "Flora" database of plant community descriptions (relevés). Clusters containing dominant species and ecological conditions characteristic of a particular forest community type according to the Braun-Blanquet system were grouped into the respective category. This process involved an automated analysis of frequency tables of cluster values and their spatial overlap with relevé coordinates, as well as expert assessment for ambiguous cases. In some cases, as indicated in Lines 362-370 in revised version of manuscript, field survey points of non-forest communities fell on the pixels of the cluster model due to the poor quality of spatial referencing of relevés. Since we initially set out to work only with forest communities in this study, such relevés points were excluded a priori; however, we felt it necessary to show in the manuscript that such a scenario is also possible and can be considered a methodological feature that should be emphasized.

Concerning the use of Random Forest for validation, you are absolutely right, and we are willing to expand the corresponding section in the methodology. Random Forest was employed in a hybrid approach to assess the contribution of each vegetation index and its statistical metrics to the final forest community classification. The model was trained in a supervised learning mode, where the predictors were the indices, and the target variable was the vegetation classes according to Braun-Blanquet. This allowed us to quantify the importance of each parameter for the final classification thanks to the built-in Feature Importance property, which allows us to rank the model parameters included as predictors by their importance in assigning a particular pixel to a particular forest community class. It is important to note that this approach is used exclusively for ranking model predictors, although theoretically supervised classification could be used to directly classify predictors into target classes, which leads us smoothly to the issue of reproducibility of results raised by the reviewer. A more detailed description has been added to the methods section – Line 288-321

We also acknowledge your point regarding the necessity for manual adjustments to enhance the method's applicability. Indeed, a certain degree of manual expert analysis was required during the interpretation of clusters and their assignment to specific forest community types. In the future, we plan to develop an automated classification system based on machine learning to minimize this step. However, it is already possible for researchers reproducing the results of the study in the case of similar forest community composition thanks to the developed reclassification matrix (Table 2). The technology and the composition of the predictor layers will allow the proposed methodology to be used with minor modifications, provided that appropriate relevés for verification are available.

We hope that these comments and revisions have helped to strengthen the manuscript and address the concerns raised by the reviewer. Please do not hesitate to request further clarifications or suggestions as needed.

We appreciate the opportunity to improve our work and look forward to incorporating the valuable feedback provided.

Sincerely,
Authors.

Reviewer 2 Report

Comments and Suggestions for Authors

General remarks

The manuscript  “Forest communities spatial modelling using machine learning and remote sensing data” explores the use of unsupervised learning techniques for mapping forest communities in Tatarstan. The resulting classes can be linked to the Braun-Blanquet classification system and thereby it appears possible to link a data-driven classification to actual forest communities with a specific species composition and ecological characteristics. The additional use of a feature analysis to link each community to certain characteristics is an interesting way of approaching this analysis. Despite having enjoyed the reviewing of this paper, there are some remarks/questions for the authors that I would like to see resolved before the publication of the paper.

-            While the introduction gives a very broad description of the mapping of vegetation and forest communities, I would have liked to read a more direct reference to recent studies on the topic… How did they approach this problem? Which satellite imagery was used? Which clustering algorithm?

-            It appears that the connection between the clustering classes and the braun-blanquet (BB) classification system  is mainly made based on expert judgement, but I would like to see figure 5 replaced by figure that shows better how often a model class occurs within a BB class compared to the total occurrence of that model class. This will make it easier to evaluate how well the model class overlaps with the BB classification.

-            Related to writing style: The paper is written in a very prosaic manner, resulting in a very lengthy manuscript. I believe the paper would benefit from more factual and concise writing.

Specific:

Line 37: Experience exists… à , rephrase

Line 129: The introduction to the materials and methods section is a bit extensive, this information is already clear from the introduction, doesn’t need to be repeated so detailed here

Table 1: maybe include in the appendix and give an overview of the index “groups”?

Line 230: what do you mean with “each row”?

Line 233: It is not clear to me if you end up with one label per input geometry or with multiple labels? Because I assume there are multiple pixels in one geometry?

Figure 3: I would reverse the figure à choose one zoom-in and make it bigger than the overview of the whole region. Or 2 zoom-ins of very contrasting regions.

Figure 5: can you give some extra explanation in the caption? What exactly is shown on the y-axis? The label says raster-value, which I assume to be the raster model class? But why would you show this with a boxplot? A histogram would be more interesting since raster model classes are discrete nominal classes where order is not relevant. à taking the average isn’t so informative in this case

Author Response

Dear Reviewer,

Thank you for the insightful comments and constructive feedback on our manuscript. We appreciate the time and effort invested in providing a thorough review, as it helps us enhance the quality and clarity of our work. Please allow us to address the points raised:

– While the introduction gives a very broad description of the mapping of vegetation and forest communities, I would have liked to read a more direct reference to recent studies on the topic… How did they approach this problem? Which satellite imagery was used? Which clustering algorithm?

Regarding the introduction, we acknowledge the need for more direct references to recent studies on the topic of mapping forest communities using remote sensing data and clustering algorithms. We have included a more focused literature review, highlighting the approaches, satellite imagery, and clustering techniques employed in similar studies.

– It appears that the connection between the clustering classes and the Braun-Blanquet (BB) classification system  is mainly made based on expert judgement, but I would like to see figure 5 replaced by figure that shows better how often a model class occurs within a BB class compared to the total occurrence of that model class. This will make it easier to evaluate how well the model class overlaps with the BB classification.

From Specific: Figure 5: can you give some extra explanation in the caption? What exactly is shown on the y-axis? The label says raster-value, which I assume to be the raster model class? But why would you show this with a boxplot? A histogram would be more interesting since raster model classes are discrete nominal classes where order is not relevant. à taking the average isn’t so informative in this case

Figure 5: Thank you for the thoughtful comment regarding Figure 5. You raised a valid point about the appropriateness of using a boxplot to represent discrete raster model classes. In this figure, the y-axis represents plant community classes according to the Braun-Blanquet classification based on field survey data. While a histogram or alternative visualization technique might seem more intuitive for discrete classes, we deliberately chose to present the data using boxplots for the following reasons:

  1. Frequency distribution: The boxplots effectively convey the frequency distribution of each raster value, showcasing the spread, outliers, and relative abundance of pixels associated with each class. This information is crucial for understanding the prevalence and spatial extent of different forest community types within the study area.
  2. Aggregation and association: The primary objective of this figure is to illustrate the process of aggregating individual raster values into broader forest community classes based on their correspondence with field data and the Braun-Blanquet classification system. The boxplots facilitate the visual association of raster values with their respective community types, as indicated by the labeling on the y-axis.
  3. Continuous representation: Although the raster model classes are discrete, the underlying ecological and environmental gradients that influence the distribution of forest communities are often continuous. The boxplots capture this continuous nature, allowing for a more nuanced representation of the data and facilitating the interpretation of ecological patterns.

While a histogram or other discrete visualization techniques may be more suitable for certain applications, we believe that the boxplot representation provides a comprehensive and informative overview of the data distribution, enabling a clear understanding of the aggregation process and the relationships between raster values and forest community types.

– Related to writing style: The paper is written in a very prosaic manner, resulting in a very lengthy manuscript. I believe the paper would benefit from more factual and concise writing.

We appreciate feedback on the writing style. However, we have received suggestions to use a compromise style that would appeal to a broader audience while still being a scientific text, rather than a strict scientific, fact-only style.

Specific comments:

Line 37: Experience exists… à , rephrase

We rephrase the sentence for better clarity.

Line 129: The introduction to the materials and methods section is a bit extensive, this information is already clear from the introduction, doesn’t need to be repeated so detailed here

We agree that the introduction to the materials and methods section is a bit extensive. We streamline this section to avoid unnecessary repetition of information already covered in the introduction.

Table 1. maybe include in the appendix and give an overview of the index “groups”?

We believe that presenting vegetation indices in this way allows for a better understanding of which ecological factor a particular index represents, since even within the same group indices may cover different ecological conditions and specifying only groups would not be representative.

Line 230: what do you mean with “each row”?

Due to the fact that clustering was performed on almost 300 predictors (indexes and their statistical metrics), this process became very resource-intensive even for Google Earth Engine. Therefore, an approach was used where we divided the study area into a number of smaller areas. Geometry in GIS is associated to an attribute database. Therefore, we wrote that the clustering model was applied to each line, referring to each smaller part of the study area. This wording has been clarified to avoid ambiguity.

Line 233: It is not clear to me if you end up with one label per input geometry or with multiple labels? Because I assume there are multiple pixels in one geometry?

Our approach allows for one label (or cluster assignment) per input pixel. Geometry is used here to optimize the process of applying the cluster model, as we wrote a bit above in our response to the reviewer. In general, however, it is worth noting that both the final product and the intermediate models in our study are represented by rasters rather than vectors; accordingly, we work with and analyze data sets rather than geometries.

Figure 3: I would reverse the figure à choose one zoom-in and make it bigger than the overview of the whole region. Or 2 zoom-ins of very contrasting regions.

We appreciate the suggestion to reverse the figure and prioritize the zoom-in view(s). We consider providing one or two zoom-in examples to better showcase contrasting regions, in addition to the overview.

We greatly appreciate your insightful comments and the opportunity to improve our manuscript. We carefully consider and address each of the points raised to enhance the quality, clarity, and overall presentation of our work.

Sincerely,

Authors.